# Habitat quality, configuration and context effects on roe deer fecundity across a forested landscape mosaic

**Valentina Zini** [1]*, **Kristin Wäber** [1,2], **Paul M. Dolman** [1]

**1** School of Environmental Sciences, University of East Anglia, Norwich, United Kingdom, **2** Forestry Commission, East England, Santon Downham, Brandon, United Kingdom

* v.zini@uea.ac.uk

## Abstract

Effective landscape-scale management of source-sink deer populations will be strengthened by understanding whether local variation in habitat quality drives heterogeneity in productivity. We related female roe deer *Capreolus capreolus* fecundity and body mass to habitat composition and landscape context, separately for adults and yearlings, using multi-model inference (MMI) applied to a large sample of individuals (yearlings: fecundity = 202, body mass = 395; adults: fecundity = 908, body mass = 1669) culled during 2002–2015 from an extensive (195 km$^2$) heterogeneous forest landscape. Adults were heavier (inter-quartile, IQ, effect size = +0.5kg) when culled in buffers comprising more arable lands while contrary to our prediction no effects on body mass of grassland, young forest or access to vegetation on calcareous soil were found. Heavier adults were more fertile (IQ effect size, +12% probability of having two embryos instead of one or zero). Counter-intuitively, adults with greater access to arable lands were less fecund (IQ effect of arable: -7% probability of having two embryos, instead of one or zero), and even accounting for greater body mass of adults with access to arable, their modelled fecundity was similar to or lower than that of adults in the forest interior. In contrast, effects of grassland, young forest and calcareous soil did not receive support. Yearling body mass had an effect on fecundity twice that found in adults (+23% probability of having one additional embryo), but yearling body mass and fecundity were not affected by any candidate habitat or landscape variables. Effect of arable lands on body mass and fecundity were small, with little variance explained (Coefficient of Variation of predicted fecundity across forest sub-regions = 0.03 for adults). More variance in fecundity was attributed to other differences between forest management sub-regions (modelled as random effects), suggesting other factors might be important. When analysing source-sink population dynamics to support management, an average value of fecundity can be appropriate across a heterogeneous forest landscape.

**Data Availability Statement:** Data are available under the Freedom of Information Act 2000 (contact via Andrew Stringer Andrew. stringer@forestryengland.uk) Data on arable lands and grasslands were derived from the Land Cover Map 2007 which is available from the Centre for

Ecology & Hydrology (https://www.ceh.ac.uk); data upon request via online Quote Request Form https://www.ceh.ac.uk/services/land-cover-map-2015).

**Funding:** PDM award, grant number R203625. This study was funded by Forestry Commission England (East England Forest District). Data collection was carried on by Forestry Commission rangers.

**Competing interests:** The authors have declared that no competing interests exist.

## Introduction

Deer populations are increasing in both North America and Europe [1–3], with important consequences for biodiversity, human health and traffic collisions [4–6]. Management effectiveness is improved by landscape-scale analysis of source-sink demography [7]; however, measuring the required parameters of density, fecundity, neonatal and adult mortality can be time consuming and challenging [7,8]. Local, context-specific, measures are required as deer fecundity can vary with both density [9,10] and landscape suitability [11–13]. Although fecundity can be assessed where culled carcasses are available [7], it is important to understand the degree to which fecundity varies across a heterogeneous landscape and the scale at which management can be simplified by spatially-averaging data.

Roe deer *Capreolus capreolus* fecundity and its relation with body mass vary substantially between populations. Rates of implantation failure vary across Britain, in relation to climatic severity, whilst within populations there is no consistent effect of weather on female fecundity [14]. Similarly, fertility (the percentage of pregnant 2-year-old females) was lower in a Norwegian than in a French roe population of similar density, despite greater body mass [15]. In Britain, although fertility was positively related to body mass overall, the asymptote at which fertility became maximal differed among 15 populations [16]. Fertility also varies within a population in relation to local landscape heterogeneity. Female roe deer with greater availability of preferred habitat within their winter home-range had larger litters in the subsequent spring [13]. In an enclosed roe deer population in Northern France, female lifetime reproductive success was positively associated with presence of open habitat edges within the home range and negatively associated with mature open forest [12]. Similarly, in central Japan [11] GPS-collared sika deer (*Cervus nippon*) with greater access to forest edges had higher probability of pregnancy. Climate, density and food availability also influence red deer (*Cervus elaphus*) fertility [17].

We examined the degree to which forest habitat (stand structure and soil type) and landscape context (access to preferred non-forest habitats) affect body mass and fecundity of roe deer. We used extensive cull data collected from 2064 individuals, in a landscape-scale (195 km$^2$ forest mosaic in eastern England), long-term (14 year) study. Tree crop, forest growth stage and soil composition vary locally and across the extensive forest landscape; while the progression of annual felling-replanting activity across forest sub-regions further de-coupled growth stage composition from local soil, forest configuration and landscape context. Together, this provided an unusual degree of landscape replication allowing powerful analysis.

We predicted that individuals with better quality habitat would have greater body mass and fecundity. Roe deer prefer early-successional relative to mature forest habitats [18,19], likely as these provide both concealment and high quality forage [20–22]. Similarly, roe in the study area preferentially utilise young stands (0–10 years) prior to canopy closure, where ground vegetation, particularly bramble *Rubus fruticosa* agg., are available [23,24]. Calcareous soil supports greater understory plant species richness [25], thus we hypothesized better quality home ranges to have greater local availability of young forests, calcareous soils, grasslands, and arable lands. Furthermore, we predicted that habitat effects would differ between yearling and adult females due to greater sensitivity of yearling fecundity to both habitat and body mass [26].

## Material and methods

### Study area

The study was conducted in Thetford Forest (Fig 1), a conifer-dominated lowland plantation, established during the 1930-1950s on sandy soils in eastern England. Conifer crops (see S1 File

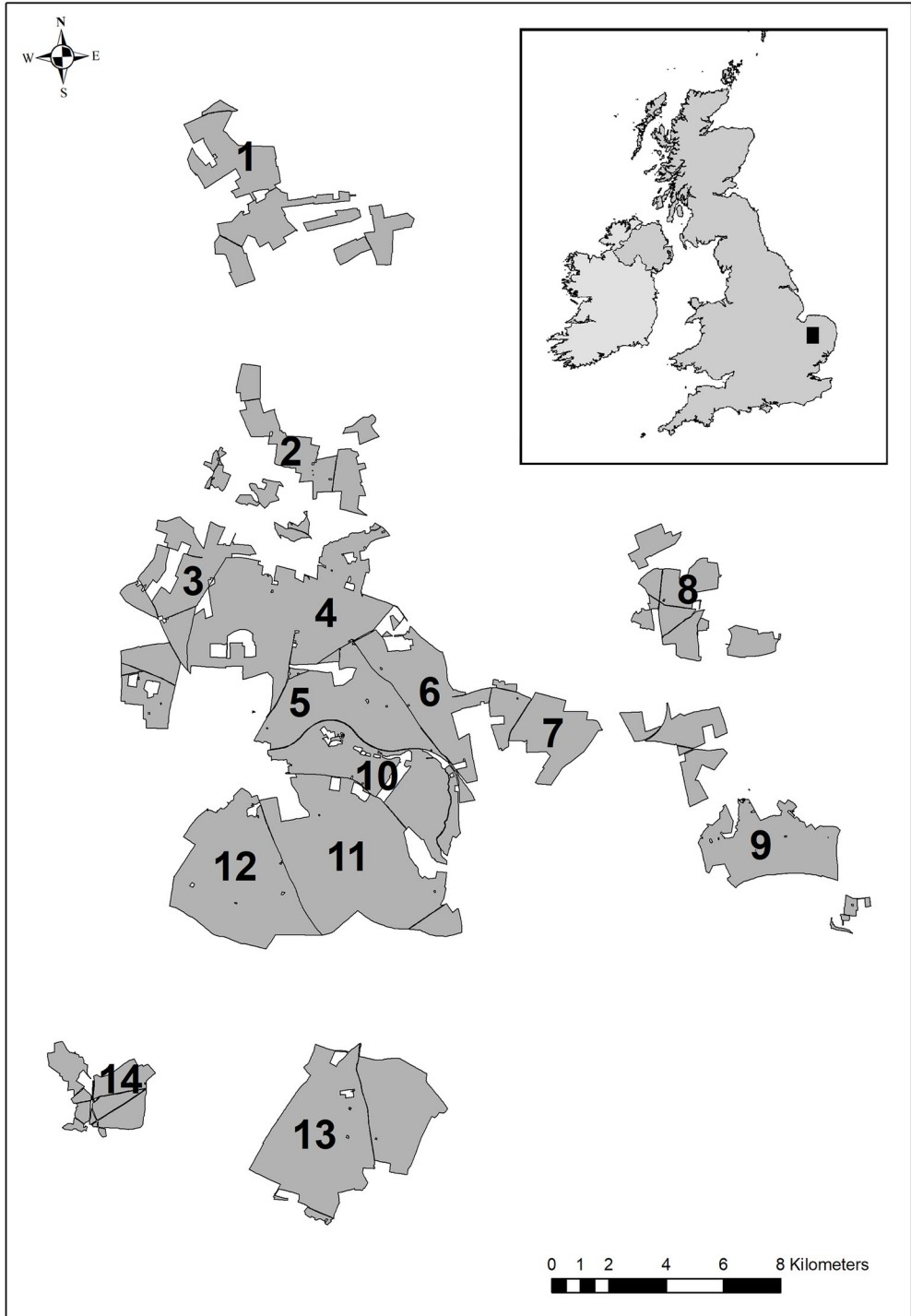

**Fig 1. Study area map.** Thetford Forest showing management sub-regions (see S1 File).

for composition) managed by clear-felling and replanting at economic maturity comprised a mosaic of discrete even-aged plantings (hereafter 'coupes'; mean area = 9.0 ha ± 8.6 SD). We sub-divided the forest landscape into 14 'sub-regions' (mean area = 13.2 km², SD = 5.6, Fig 1), that varied in their proportionate representation of deciduous plantings, soil types ('acidic'

podsols and gravelly sands; 'calcareous' chalky sands and rendzinas) and configuration (S1 File). Sub-region's area, perimeter, perimeter-area ratio, and percentage of calcareous soil were independent, with replicates for each relative combination of factors (S1 File). This permitted effects of habitat and landscape composition to be estimated without confounding geographical effects.

Four deer species were present in the study area [7]: re-established native roe and red, naturalised fallow *Dama dama*, and introduced Reeves's muntjac *Muntiacus reevesi*. In the absence of large predators, deer populations are managed by wildlife rangers employed by Forestry Commission England (FC) to mitigate their impacts on forest crop establishment, biodiversity and road collisions. Forest-wide density of roe deer (estimated using nocturnal distance-sampling by thermal imaging [8]) remained low throughout the study period, fluctuating between ca. 2.6 to 4.7 deer/km$^2$ (Zini et al., unpublished data).

## Fecundity and body mass data

Female roe body mass and fecundity data for 2002–2015 were obtained through a long-term research collaboration with FC. For each individual, the cull location (recorded using hand-held GPS), date, sex, 'body mass' (measured to the nearest 0.1 kg, after removal of head, feet and viscera, hung with blood drained), number of embryos and remarks (e.g. if shot damage affected body mass) were recorded. Age was estimated by tooth eruption [27] a method that reliably classes individuals as juvenile (born the preceding summer), yearling (in their second winter) or adult; it is however worth noting that a small error may occur when classifying yearlings [28]. Carcasses recorded in the database as damaged (i.e. incomplete) or considered likely to be damaged as they weighed less than a threshold of 6 kg for yearlings and 8 kg for adults (see S2 File), were excluded from analysis.

Roe deer exhibit delayed implantation [29] and in Britain embryos are not visible on opening the uterus before early January [16]. We used data from 1$^{st}$ January to 31$^{st}$ of March and included week in models as a fixed effect to account for the seasonal increase in embryo detectability (for further detail, see S3 File). We couldn't account for any senescence effects, however it is likely that our heavily hunted, low density population comprised only a small fraction of senescent-aged roe deer. We therefore assumed effects of ageing [14] to be negligible and non-confounding.

## Environmental variables

We selected four *a priori* habitat variables considered likely to affect roe deer performance: calcareous soil, young forest, arable lands and grasslands (Table 1).

Land cover within Thetford forest was extracted from the FC GIS management database that provided spatially-explicit information on the area, crop species and planting year of each management polygon (mean = 0.029 km$^2$ SD = 0.03 km$^2$). Surrounding grasslands and arable lands were extracted from the Land Cover Map 2007 (hereafter LCM 2007; [30]) that maps 23 classes to 0.5 ha resolution. All LCM 2007 grassland and heathland classes were combined as 'grassland' as finer ecological categories were not reliably classified (S4 File). Permanent unplanted open space within Thetford forest was also classified as grassland. We considered the percentage of young forest combining felled and unplanted, recently restocked and pre-thicket growth stages aged 0–10 years) in each culling year. The percentage of calcareous soil was calculated for the forest and grassland area, excluding arable that provides high-quality food (in this region comprising wheat, barley, oilseed rape, potatoes and maize) irrespective of underlying soil class. See S4 File for details of soil classification and data. Variables were

**Table 1. Candidate environmental variables.**

| Variable | | | Adult | | Yearling | |
|---|---|---|---|---|---|---|
| | | | body mass | fecundity | body mass | fecundity |
| %Young forest | Mean | | 19 | 16 | 15 | 16 |
| | CV | | 1.1 | 1.2 | 1.2 | 1.2 |
| %Arable | Mean | | 19 | 14 | 15 | 14 |
| | CV | | 1.0 | 1.3 | 1.3 | 1.3 |
| %Grassland | Mean | | 10 | 8 | 7.8 | 8 |
| | CV | | 1.1 | 1.5 | 1.4 | 1.5 |
| %Calcareous soil | Mean | | 38 | 40 | 41 | 40 |
| | CV | | 0.7 | 0.8 | 0.8 | 0.8 |

Variables examined in models of adult and yearling female roe deer fecundity and body mass, showing their mean and coefficient of variation (CV, proportion of SD relative to mean extracted at the Aikike-weighted mean buffer radius).

extracted from GIS layers using R statistical software [31] and the packages "sp" [32], "rgeos" and "rgdal" [33].

## Extracting environmental data at scales relevant to roe deer home range

Separately, we sought to relate individual body mass and fecundity, to habitat quality in the area readily available to that individual. Adult roe deer have been shown to be sedentary within a defined home-range [34], particularly in autumn and winter (e.g. [35]). As deer were managed by stalking rather than drive hunting, we assumed each individual was shot somewhere within its home-range. Given uncertainty in home-range extent (and in the absence of telemetry data for culled individuals), rather than assuming a single *a priori* home-range size we ran a series of models that related fecundity and body mass to habitat variables extracted from biologically-plausible buffer radii around the individual's cull location. Based on the monthly home-range extent reported in a previous study in Thetford Forest [36] and other western European study sites with similar temperate climate (northern Italy, France and Germany, see [37]), our buffer radii ranged from 400m to 600m (59ha to 113ha) with 50m increments. Models at different radii were subsequently weighted by their predictive ability (see below).

## Analyses

Body mass was analysed by Generalised Linear Mixed Models (GLMMs) with random intercept and normal error distribution. Body mass models incorporated random effects of forest sub-region, to account for reduced spatial independence of individuals culled from nearby areas and variation between subregions in roe density, human recreational density and other factors. However subregions were larger than finer-grained variation in local density or recreation so only partly account for these effects. Models also incorporated random effects of cull year (to control for severity of the winter in which the individual was culled, plus any potential lagged weather-related effects of variation in forage availability from the previous growing season) and calendar week (as a continuous variable to control for seasonal body mass fluctuations).

Adult fecundity examined whether the female carried twins (two embryos), versus zero or one embryos (combined as the reference level), given the high frequency of twins (63%) and low frequency of non-pregnant females (8.5%), in GLMMs with binomial error. Fecundity of yearlings (that were born two summers before the winter in which culled, aged 20–22 months at culling, and approximately 15 months when first impregnated), was analysed by ordinal

logistic models, as 20% were non-pregnant, 54% carried one embryo and 26% carried twins, using R package "ordinal". Fecundity was modelled incorporating landscape and environmental variables and the individual's body mass, to interpret direct environmental effects from indirect effects acting via body mass. All fecundity models included a fixed effect of calendar week to account for embryo detectability (coded as a categorical dummy variable, adults: 0 = weeks 1–3, 1 = weeks 4–12; yearlings: 0 = weeks 1–2, 1 = weeks 3–12; see S3 File) and random effects of forest sub-region and cull year.

Model selection and parameter estimation were performed using Multi-Model Inference (MMI) in an information theoretic framework [38] using 'MuMIn' package in R [39]. For each incremental buffer radius we built candidate models comprising all possible variable combinations, we then examined the relative performance of the full complement of candidate models across the range of radii, with parameters averaged across the 95% model confidence set, weighting models by their Akaike weight. Environmental variables were considered supported if included in the 95% model confidence set and their model-averaged parameter CI did not span zero. Explanatory power was assessed as the $R^2$ of models comprising supported variables only at the Aikike-weighted mean buffer radius.

Correlation between explanatory variables was lower (maximum r from pairwise comparisons between all explanatory variables: 0.27) than the threshold ($r > 0.78$) of confounding intercorrelation [40]. Spatial autocorrelation of residuals of reduced models incorporating supported variables only was examined using Moran's I (Package "spdep", [41]), defining neighbour objects as the GPS points occurring inside the Aikike-weighted mean buffer radii and assigning spatial weights to the neighbouring GPS points using row standardization (see [42]). Consequences of environmental variables for fecundity were examined from model-averaging with variables extracted at the Aikike-weighted mean buffer radius, first by solely considering the direct effect of environment on fecundity (not including body mass in the model), second by holding body mass at the forest-wide mean, and third also accounting for the indirect effect of environment on fecundity as mediated by body mass (with body mass resampled from the body mass—environment model, according to model mean and standard error, and incorporated in the fecundity—environment model, to predict the overall distribution of fecundity with environment).

Last, we examined the degree to which variation in habitat and landscape context caused fecundity to vary between the 14 forest sub-regions. The spatial distribution of adult and yearling body mass was predicted for each cell of a raster of 100m resolution across the study area, using a reduced model including supported variables only, extracted from the Aikike-weighted mean buffer radius built around the centroid of each 100m x 100m raster cell and then averaged across cells in each forest sub-region. Mean fecundity of each forest sub-region and the variance between them was then predicted from the raster of predicted body mass, together with supported environment and landscape variables.

All analyses were performed in R 3.2.5 [31].

## Results

### Body mass

Mean body mass of adults was 13.7 kg (SD = 1.65) and of yearlings 12.1 kg (SD = 1.71), with little variation observed between forest sub-regions ($CV_{adults}$ = 0.05 $CV_{yearlings}$ = 0.06, S7). Only a small subset of models comprised the 95% confidence set (13% of all candidate models). The Aikike-weighted mean buffer radii across the 95% confidence set of adult body mass models was 578m (SD = 27m). Adult female roe deer were heavier when culled in localities comprising a greater percentage of arable lands (Figs 2 and 3).

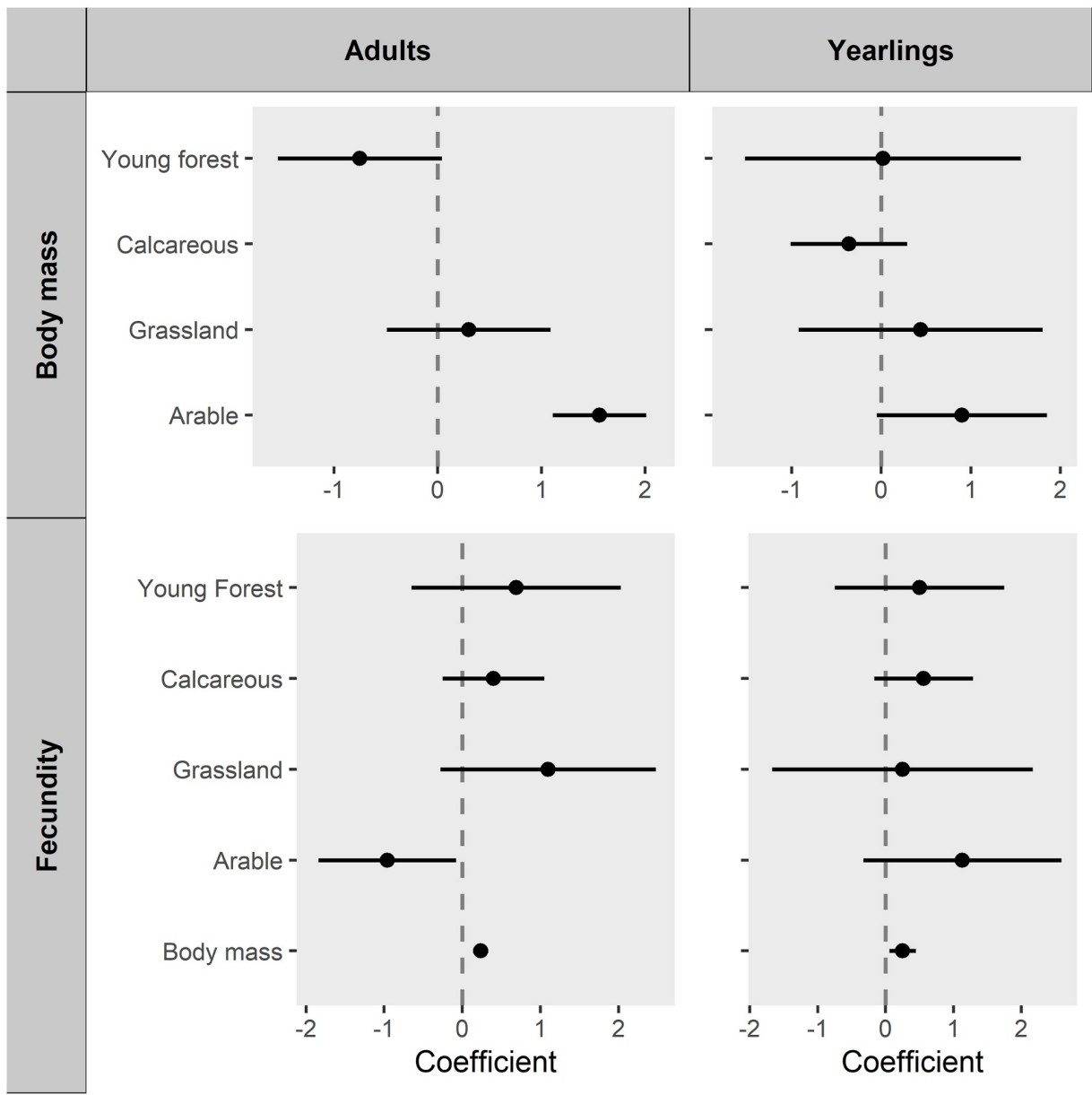

**Fig 2. Models relating adult and yearling roe deer body mass and fecundity to landscape and environment.** Coefficients of environmental variables, model-averaged across buffer radii (filled symbol) and their 95% Confidence Interval (CI, bars), are presented. All models controlled for random effects of calendar week (as a continuous variable in body mass models, as a dummy categorical variable in fecundity models), forest sub-region and culling year.

Adult body mass model had a conditional $R^2$ (weighted-average by Akaike weights across radii) of 0.13 and marginal $R^2$ of 0.03. No residual spatial autocorrelation was found in our models.

In contrast to adults, no effect of any of our environmental variables was supported for yearlings' body mass (Fig 2). The subset of candidate models comprising the 95% confidence set was 80% of all candidate yearlings body mass models. The Aikike-weighted mean buffer radii, across the 95% confidence set, was 500m (SD = 76m).

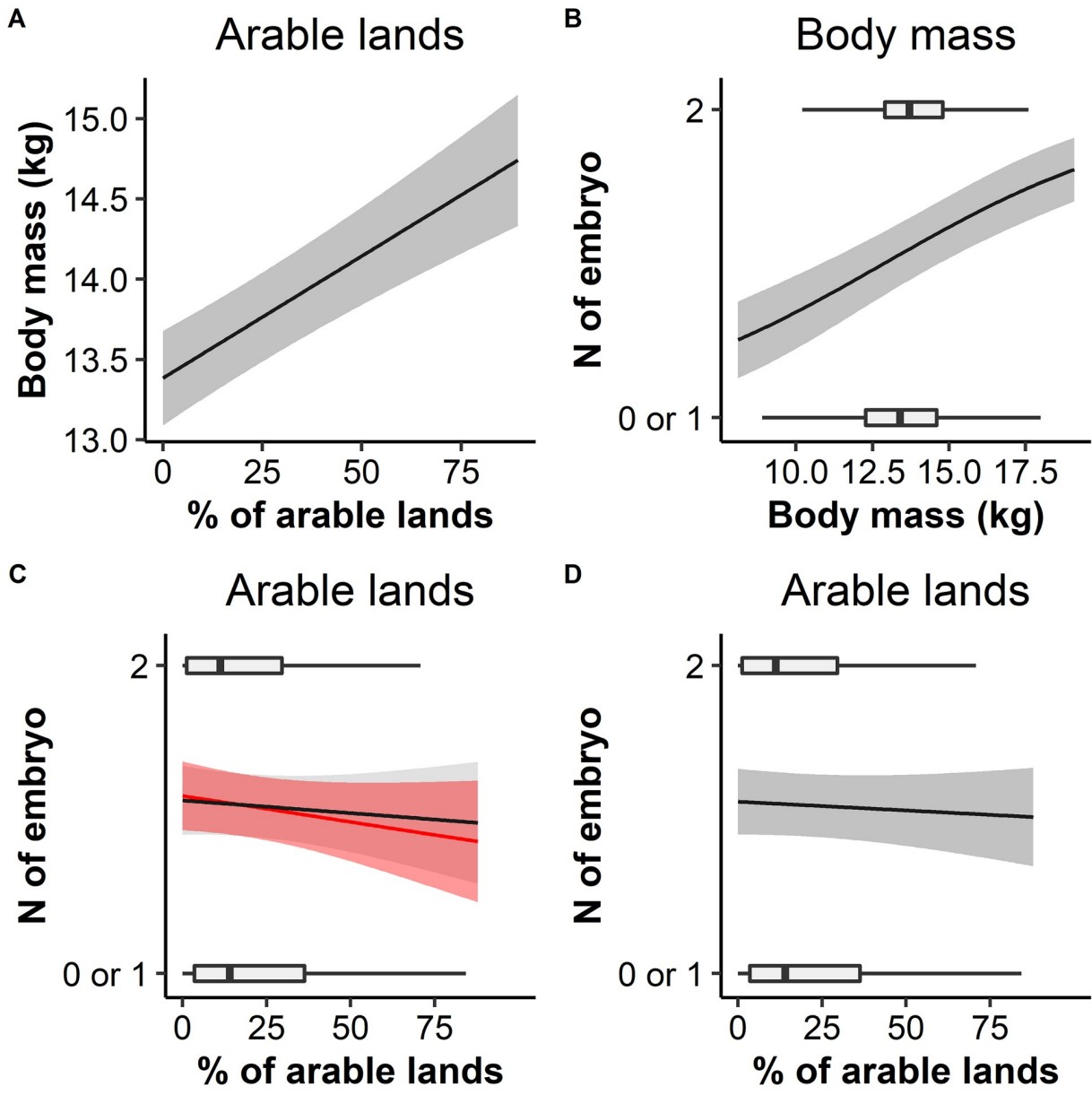

**Fig 3. Predicted adult roe deer body mass and fecundity in relation to percentage of arable lands.** (A) Body mass is predicted from model-averaging the body mass-environment model, with variables measured at the Aikike-weighted mean buffer radius, controlling for random effect of calendar week, cull year and forest sub-region. Fecundity is predicted from model-averaging the fecundity-environment model, with variables measured at the Aikike-weighted mean buffer radius, incorporating calendar week (to account for embryo detectability) and random effects of cull year and forest sub-region. (B) shows the relation of fecundity to body mass, holding other variables at their mean; (C) shows the relation of fecundity to % arable, predicted when holding body mass at the forest-wide mean (red line) and accounting for greater body mass of adults with more arable (with body mass resampled according to arable extent, from the body mass–environment model) (black line); (D) shows the direct relation of fecundity to % arable predicted from a model including % of arable lands only.

### Fecundity

Mean fecundity calculated from models controlling for calendar week was 1.6 (SD = 0.5) embryos per adult female (mean fecundity at maximum embryo detectability weeks 4–12 = 1.55) and 0.9 (SD = 0.7) embryos per yearling female (mean fecundity at maximum embryo detectability, week 3–12 = 1.0).

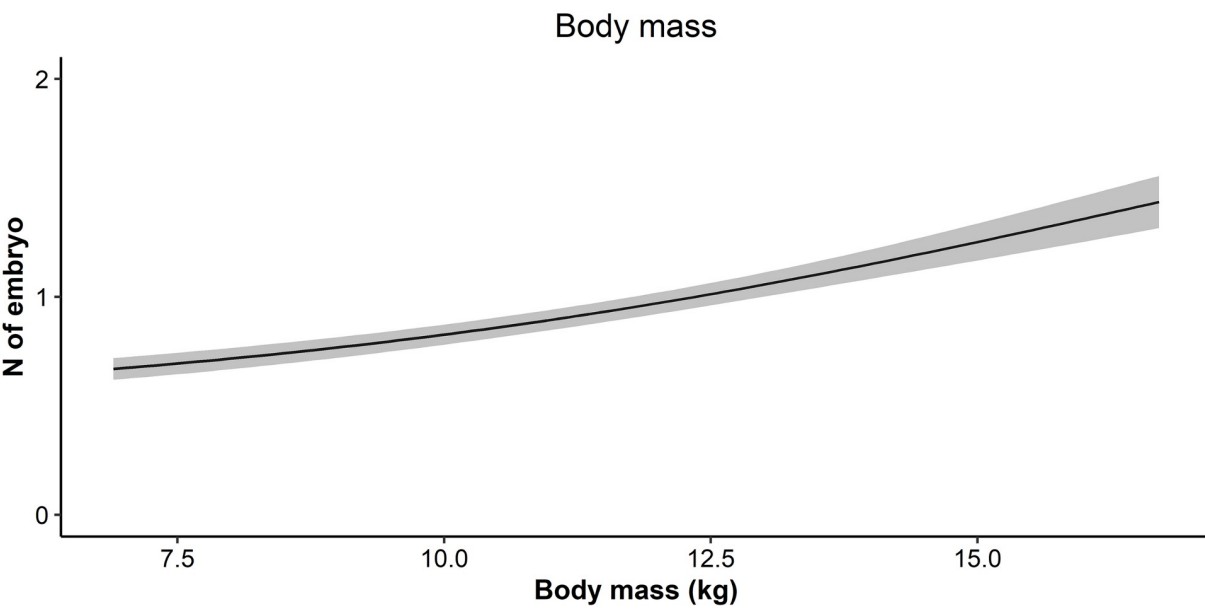

**Fig 4. Predicted yearling fecundity in relation to body mass.** Fecundity predicted from the averaged-model incorporating body mass and calendar week to account for embryo detectability, and random effects of cull year and forest sub-region.

The subset of candidate models comprising the 95% confidence set was 38% of all candidate adult fecundity models. The Aikike-weighted mean buffer radii, across the 95% confidence set, was 500m (SD = 72m). Adults were more fecund when heavier (Figs 2 and 3B). The response of adult fecundity to the farmland boundary was complex and counter-intuitive. Model-averaging showed fecundity was reduced by a greater percentage of arable lands (fixed additive effect Figs 2 and 3C) and holding body mass at the forest-wide mean, adults were less fecund close to the arable boundary (Fig 3C, red line). However, fecundity was more strongly affected by body mass than arable lands (Fig 2) and body mass was greater at the arable boundary (Fig 3A). Accounting for both the direct effect of arable on fecundity, and indirect effect mediated by the relation of body mass to arable, overall fecundity of adult females close to farmland was similar to or lower than that of females in the forest interior (Fig 3C black line). No support was found for an effect of grassland, young forest or calcareous soils on adult fecundity. Adult fecundity models had a mean conditional $R^2$ (weighted-average by Akaike weights across radii) of 0.17 and marginal $R^2$ of 0.07.

The subset of candidate models comprising the 95% confidence set was 64% of all candidate yearling fecundity models. The Aikike-weighted mean buffer radii, across the 95% confidence set, was 500m (SD = 102m). Yearlings also were more fertile when heavier (Figs 2 and 4), but no effect of any environmental variable was supported (Fig 2).

Average adult fecundity calculated from predictions (including predicted spatial distribution of body mass) was 1.5 (SD = 0.1) embryos, similar to the raw means. Mean predicted fecundity per sub-region ranged from 1.5 to 1.6 for adults, with low variance between sub-regions ($CV_{Adults}$ = 0.03; S1 Table).

## Discussion

This study utilised a large sample of individuals collected across an extensive landscape mosaic over 14 years. Only arable lands contributed to observed variation in fecundity and body mass, but with both expected and counter-intuitive effects.

Roe is a medium-sized deer with a 'concentrate-selector' feeding strategy [43], depending on relatively high quality foods [22,44]; their weight is expected to increase with better habitat quality. Adult roe deer were slightly heavier (inter-quartile, effect size = +0.5kg) when culled in areas with a greater percentage of arable lands. Consistent with our findings, a previous study [45] showed that roe deer in fragmented woodlands in an arable landscape were heavier and fecal samples in these areas had higher levels of nitrogen and phosphorous compared to roe in a forest environment, implicating higher nutritional content of arable forage as contributing to weight. Contrary to our predictions, no effects of calcareous soil, grasslands or young forest was found on roe body mass. This is possibly due to the average body mass being high already so food limitation may be negligible.

Adult female roe deer culled in localities buffered by a greater percentage of arable lands were heavier, conferring reproductive advantage. However, this was offset by a negative effect of arable area on fecundity (inter-quartile effect size of arable: -7% probability of having two embryos, instead of one or zero), so that fecundity of adult females at the farmland boundary was similar to or less than those in the forest interior, even accounting for the positive indirect effect of arable on fecundity mediated by body mass. Fecundity data were collected during late winter, when shortage of food determines fecundity. Roe deer graze on arable lands particularly during winter [46] when young crops or weedy stubbles are widespread; so the lower than expected fecundity of individuals with greater access to arable was counter-intuitive, and contradicted the 'fragmentation nutrition hypothesis' [47], that proposes fragmented woodlands in a matrix of nutritious farmland favour deer population growth. Within the forest, muntjac and roe deer densities are greater closer (<350 m) to arable lands (Zini et al unpublished data). It is possible that fecundity, and apparent habitat 'suitability', may be reduced by intra and inter-specific agonistic interference and stress [2] close to the forest-arable margin. Fecundity wasn't related to the percentage of young forest, grasslands or calcareous soil.

Yearling body mass and fecundity were not related to any environmental variables tested. Yearling performance is likely to be influenced by their natal (i.e. their mother's) home range, where most growth takes place within the first year [48]. However depending on body condition, some individual are philopatric while approximately a third undertake natal dispersal away from the maternal home-range at 10–12 months of age [49], making it more difficult to detect a relationship between the newly established home range (where they were culled at 20–23 months of age) and their performance. In contrast adults are more likely to be occupying an established home range when culled.

In this study, heavier adult females were slightly more fertile (inter-quartile effect size of weight: +12% probability of having two embryos, instead of one or zero), as widely documented by previous studies [14,26]. Interestingly, for adult fecundity the effect of body mass was greater, but with the same order of magnitude to that of arable lands, while the effect of body mass on yearling fecundity was twice (+23% probability of having one additional embryo) that found for adults. In roe deer, body mass plays a predominant role in determining potential litter size at the conception stage [26], determining the maximum reproductive output for a given body size; therefore having a greater influence on yearlings that–in contrast to adults–haven't yet reached their maximum body size [48].

Although ecologically interesting, the small overall effect size of landscape context (proximity to arable lands) resulted in little variation in predicted (or observed) fecundity among forest sub-regions, while none of the other factors considered in this study affected roe deer performance. Our study suggests it will be sufficient to base estimates of population fecundity on forest-wide measures.

## Supporting information

**S1 File. Study landscape.**
(DOCX)

**S2 File. Validation of the larder dataset.**
(DOCX)

**S3 File. Effect of date on detectability of embryos.**
(DOCX)

**S4 File. Land cover and soil data.**
(DOCX)

**S1 Table. Predicted fecundity per forest subregion.** Predicted fecundity of adult roe deer calculated as an unweighted average across raster cells within each of 14 forest subregions from environmental fecundity models and coefficient of variation of forest subregion's predicted fecundity.
(DOCX)

## Acknowledgments

We are grateful to the wildlife ranger team of Thetford Forest, particularly B. Ball, D. Gunn, S. Hetherington, P. Mason, and T. Parr. FC provided cull data from 2002–15. We are grateful to Mark Hewison and an anonymous referee for helpful comments on an earlier version of this paper.

## Author Contributions

**Conceptualization:** Paul M. Dolman.

**Data curation:** Valentina Zini, Kristin Wäber.

**Formal analysis:** Valentina Zini.

**Funding acquisition:** Paul M. Dolman.

**Methodology:** Paul M. Dolman.

**Supervision:** Paul M. Dolman.

**Writing – original draft:** Valentina Zini.

**Writing – review & editing:** Valentina Zini, Kristin Wäber, Paul M. Dolman.

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
