## [Decision Letter · Decision Letter 0]

17 Oct 2019

PONE-D-19-25714

Habitat quality, configuration and context effects on roe deer fecundity across a forested landscape mosaic

PLOS ONE

Dear Ms Zini,

Thank you for submitting your manuscript to PLOS ONE. After careful consideration, we feel that it has merit but does not fully meet PLOS ONE’s publication criteria as it currently stands. Therefore, we invite you to submit a revised version of the manuscript that addresses the points raised during the review process.

The two reviewers agree that the manuscript is almost ready for publication. Reviewer 2 still asks for a novel plot of the relationship between fecundity and %arable land based on raw data and not model predictions. I agree that this would be helpful for the reader. I also encourage the authors to take into account the other minor recommendations made by Reviewer 2. Once this is done, and whatever the plot based on raw data reveals, I am ready to accept your manuscript without sending it for reviews again (provided that your interpretation is adequately articulated with those novel results).

We would appreciate receiving your revised manuscript by Dec 01 2019 11:59PM. To enhance the reproducibility of your results, we recommend that if applicable you deposit your laboratory protocols in protocols.io, where a protocol can be assigned its own identifier (DOI) such that it can be cited independently in the future. For instructions see: http://journals.plos.org/plosone/s/submission-guidelines#loc-laboratory-protocols

We look forward to receiving your revised manuscript.

Kind regards,

Franck Jabot

Academic Editor

PLOS ONE

Journal Requirements:

3. We note that Figure 1 and Supporting Information file Figures in your submission contain map/satellite images which may be copyrighted. All PLOS content is published under the Creative Commons Attribution License (CC BY 4.0), which means that the manuscript, images, and Supporting Information files will be freely available online, and any third party is permitted to access, download, copy, distribute, and use these materials in any way, even commercially, with proper attribution. For these reasons, we cannot publish previously copyrighted maps or satellite images created using proprietary data, such as Google software (Google Maps, Street View, and Earth). For more information, see our copyright guidelines: http://journals.plos.org/plosone/s/licenses-and-copyright.

You may seek permission from the original copyright holder of Figure(s) [#] to publish the content specifically under the CC BY 4.0 license. 

If you are unable to obtain permission from the original copyright holder to publish these figures under the CC BY 4.0 license or if the copyright holder’s requirements are incompatible with the CC BY 4.0 license, please either i) remove the figure or ii) supply a replacement figure that complies with the CC BY 4.0 license. Please check copyright information on all replacement figures and update the figure caption with source information. If applicable, please specify in the figure caption text when a figure is similar but not identical to the original image and is therefore for illustrative purposes only.

Reviewers' comments:

Reviewer's Responses to Questions

**Comments to the Author**

1. Is the manuscript technically sound, and do the data support the conclusions?

Reviewer #1: Yes

Reviewer #2: Partly

2. Has the statistical analysis been performed appropriately and rigorously? 

Reviewer #1: Yes

Reviewer #2: Yes

3. Have the authors made all data underlying the findings in their manuscript fully available?

Reviewer #1: Yes

Reviewer #2: Yes

4. Is the manuscript presented in an intelligible fashion and written in standard English?

Reviewer #1: Yes

Reviewer #2: Yes

5. Review Comments to the Author

Reviewer #1: Dear authors

Your manuscript deals with a interesting study on the effect of Habitat quality on roe deer fecundity across a forested landscape mosaic. Suggestions were exhaustively accepted and now the paper is improved and well written.

Reviewer #2: General comments

This paper analyses quite a large data set of hunted roe deer in terms of the landscape drivers of variation in performance as indexed by female body mass and reproductive parameters. This is a revised version of a manuscript that I reviewed some time ago. The authors have made considerable effort in revising the paper in relation to my comments and those of a second referee. I found that the revised version was significantly improved overall, although there are some issues that I highlighted that the authors have preferred not to account for. Because some of these issues could be seen as related to personal viewpoint, at the end of the day this is a matter of choice and does not preclude publication. I have tried to explain why I do not necessarily agree with some of these choices below, but I leave it up to the authors to judge. However, there is one major point that requires further clarification. I still cannot see an analysis of the raw data relationship between female fecundity and the proportion of arable land irrespective of body mass, i.e. I want to see an analysis (and figure) of this relationship when body mass is not included in the model. Otherwise, I have difficulty buying the interpretation that a negative relationship between fecundity and % arable land offsets the advantage obtained through higher body mass in more arable areas (see below).

Major Comment:

A. From the new figures, it is somewhat easier to get a handle on the relationships. Indeed, it is clear from the Figure 3 that the effect of arable land on body mass is quite strong and robust. In contrast, it is also clear that there is very little support for any biologically meaningful relationship between % arable land and fecundity, while controlling for body mass (Fig. 3C). Indeed, firstly, the confidence intervals for estimates of fecundity at 5% and 80% arable land clearly overlap. Hence, I do not buy the interpretation on lines 296-301 that fecundity decreases with increasing proportion of arable land. Secondly, I still do not see the equivalent analysis of the raw data relationship between fecundity and % arable land ie IGNORING body mass completely in the model. Given the strong relationship between fecundity and body mass that they show, I expect that the relationship between fecundity and % arable is also positive if you remove body mass from the fecundity model altogether. Please show this. Otherwise the interpretation that the body mass advantage of having access to arable land is offset by a negative effect on fecundity is not convincing.

Other comments

B. Concerning the management-orientated context for this work, of course I agree that information on variation in female fecundity in relation to landscape structure could have management implications. However, in almost all contexts where ungulate population dynamics have been studied to date, variation in this component is swamped by spatial and temporal variation in the early survival component. As a result, population performance is more informatively indexed by fawn survival during their first summer. Therefore I would question whether indeed “knowledge of fertility across the landscape is also necessary for deer management”. However, I recognise that this is, to a certain degree, a question of personal opinion.

C. While the authors have now provided details on how they aged the animals, despite what they say on line 107, I know of no study that has demonstrated that tooth eruption provides reliable information to distinguish yearling animals in roe deer. What is the basis for this assumption? The error is not huge for yearlings (e.g. see Hewison et al. 1999), but it should at least be acknowledged.

D. For the senescence aspect, you can make this assumption if you wish, and intuitively I expect that it is a fair assumption given the hunting pressure, but I strongly dispute that you have reliable estimates of observed early survival as you can only get this information if you mark fawns at birth. Female:fawn ratios are an unreliable proxy for this in roe deer, as it is virtually impossible to distinguish yearlings (ie non-reproductive) from adult females in autumn. Hence, the observed ratio is strongly dependent on population age structure, e.g. if performance is high, then there will be proportionately more yearlings in the population, which will artificially decrease the observed female:fawn ratio because yearlings (18 months old in autumn) have not yet reproduced. This means that a high performance population may have a lower female:fawn ratio simply because of the fact that there are more immature females in the population which are impossible to reliably distinguish and so are included in the calculation of the ratio. As I previously said, the authors should just say that they could not account for any senescence effects, so that they must assume they are negligible, which is quite likely in a heavily hunted population at low density.

E. In terms of the putative source-sink dynamics of this system, I remain unconvinced. Source-sink dynamics cannot be inferred from hunting data alone. A source sink system implies that the sink population is unable to sustain itself (r < 1) in the absence of immigration from the source population (r > 1). Just because there is variation in performance between neighbouring populations, one cannot therefore assume that the system contains a source and sink without detailed information on i/ population growth rate AND ii/ net movement of animals between the two sites. I do not see how the authors can obtain such information on movements from hunting data of non-marked animals. I therefore strongly suggest that this consideration be removed, e.g. first sentence of abstract and elsewhere.

Mark Hewison

6. PLOS authors have the option to publish the peer review history of their article (what does this mean?). If published, this will include your full peer review and any attached files.

Reviewer #1: No

Reviewer #2: Yes: A.J; Mark Hewison

---

## [Author Response · Author response to Decision Letter 0]

28 Nov 2019

Below, we copy the original comments of each reviewer (in black font), and beneath each we detail how the manuscript has been amended (in Red).

Review Comments to the Author

Reviewer #1: Dear authors

Your manuscript deals with a interesting study on the effect of Habitat quality on roe deer fecundity across a forested landscape mosaic. Suggestions were exhaustively accepted and now the paper is improved and well written.

Reviewer #2: General comments

This paper analyses quite a large data set of hunted roe deer in terms of the landscape drivers of variation in performance as indexed by female body mass and reproductive parameters. This is a revised version of a manuscript that I reviewed some time ago. The authors have made considerable effort in revising the paper in relation to my comments and those of a second referee. I found that the revised version was significantly improved overall, although there are some issues that I highlighted that the authors have preferred not to account for. Because some of these issues could be seen as related to personal viewpoint, at the end of the day this is a matter of choice and does not preclude publication. I have tried to explain why I do not necessarily agree with some of these choices below, but I leave it up to the authors to judge. However, there is one major point that requires further clarification. I still cannot see an analysis of the raw data relationship between female fecundity and the proportion of arable land irrespective of body mass, i.e. I want to see an analysis (and figure) of this relationship when body mass is not included in the model. Otherwise, I have difficulty buying the interpretation that a negative relationship between fecundity and % arable land offsets the advantage obtained through higher body mass in more arable areas (see below).

This was a very good point and we have now added a new graph showing prediction from a model that ignores body mass completely to show that there is infact a negative relationship between arable and fecundity despite the positive relationship between arable and body mass.

Major Comment:

A. From the new figures, it is somewhat easier to get a handle on the relationships. Indeed, it is clear from the Figure 3 that the effect of arable land on body mass is quite strong and robust. In contrast, it is also clear that there is very little support for any biologically meaningful relationship between % arable land and fecundity, while controlling for body mass (Fig. 3C). Indeed, firstly, the confidence intervals for estimates of fecundity at 5% and 80% arable land clearly overlap. Hence, I do not buy the interpretation on lines 296-301 that fecundity decreases with increasing proportion of arable land. Secondly, I still do not see the equivalent analysis of the raw data relationship between fecundity and % arable land ie IGNORING body mass completely in the model. Given the strong relationship between fecundity and body mass that they show, I expect that the relationship between fecundity and % arable is also positive if you remove body mass from the fecundity model altogether. Please show this. Otherwise the interpretation that the body mass advantage of having access to arable land is offset by a negative effect on fecundity is not convincing.

We have now added a new graph showing prediction from a model that ignores body mass completely

Other comments

B. Concerning the management-orientated context for this work, of course I agree that information on variation in female fecundity in relation to landscape structure could have management implications. However, in almost all contexts where ungulate population dynamics have been studied to date, variation in this component is swamped by spatial and temporal variation in the early survival component. As a result, population performance is more informatively indexed by fawn survival during their first summer. Therefore I would question whether indeed “knowledge of fertility across the landscape is also necessary for deer management”. However, I recognise that this is, to a certain degree, a question of personal opinion.

C. While the authors have now provided details on how they aged the animals, despite what they say on line 107, I know of no study that has demonstrated that tooth eruption provides reliable information to distinguish yearling animals in roe deer. What is the basis for this assumption? The error is not huge for yearlings (e.g. see Hewison et al. 1999), but it should at least be acknowledged.

We have now acknowledged the error and cited the study. 

D. For the senescence aspect, you can make this assumption if you wish, and intuitively I expect that it is a fair assumption given the hunting pressure, but I strongly dispute that you have reliable estimates of observed early survival as you can only get this information if you mark fawns at birth. Female:fawn ratios are an unreliable proxy for this in roe deer, as it is virtually impossible to distinguish yearlings (ie non-reproductive) from adult females in autumn. Hence, the observed ratio is strongly dependent on population age structure, e.g. if performance is high, then there will be proportionately more yearlings in the population, which will artificially decrease the observed female:fawn ratio because yearlings (18 months old in autumn) have not yet reproduced. This means that a high performance population may have a lower female:fawn ratio simply because of the fact that there are more immature females in the population which are impossible to reliably distinguish and so are included in the calculation of the ratio. As I previously said, the authors should just say that they could not account for any senescence effects, so that they must assume they are negligible, which is quite likely in a heavily hunted population at low density.

We have now deleted the Leslie matrix and said we couldn’t account for senescence effects. 

E. In terms of the putative source-sink dynamics of this system, I remain unconvinced. Source-sink dynamics cannot be inferred from hunting data alone. A source sink system implies that the sink population is unable to sustain itself (r < 1) in the absence of immigration from the source population (r > 1). Just because there is variation in performance between neighbouring populations, one cannot therefore assume that the system contains a source and sink without detailed information on i/ population growth rate AND ii/ net movement of animals between the two sites. I do not see how the authors can obtain such information on movements from hunting data of non-marked animals. I therefore strongly suggest that this consideration be removed, e.g. first sentence of abstract and elsewhere.

We agree that fertility is only a small part of the whole source sink-system and that other, more important informations about the system are required to infer sources and sinks. Our intention with the paper is solely to address how fertility changes throughout the landscape and whether it is or it isn’t an important factor to account for, when looking at sources and sinks (provided all the other informations are used to build sources and sinks, such as density of deer, mortality rate and juvenile survival). 

Mark Hewison

---

## [Editor Report · Decision Letter 1]

5 Dec 2019

Habitat quality, configuration and context effects on roe deer fecundity across a forested landscape mosaic

PONE-D-19-25714R1

Dear Dr. Zini,

We are pleased to inform you that your manuscript has been judged scientifically suitable for publication and will be formally accepted for publication once it complies with all outstanding technical requirements.

With kind regards,

Franck Jabot

Academic Editor

PLOS ONE
---

## [Editor Report · Acceptance letter]

12 Dec 2019

PONE-D-19-25714R1 

Habitat quality, configuration and context effects on roe deer fecundity across a forested landscape mosaic 

Dear Dr. Zini:

I am pleased to inform you that your manuscript has been deemed suitable for publication in PLOS ONE. Congratulations! Your manuscript is now with our production department. 

With kind regards,

on behalf of

Dr. Franck Jabot 

Academic Editor

PLOS ONE